# Sample Preparation and Analytical Methods for Identifying Organic Compounds in Bituminous Emissions

**DOI:** 10.3390/molecules27165068

**Published:** 2022-08-09

**Authors:** Zachary Deller, Subashani Maniam, Filippo Giustozzi

**Affiliations:** 1Applied Chemistry and Environmental Science, School of Science, STEM College, RMIT University, Melbourne, VIC 3001, Australia; 2Civil and Infrastructure, School of Engineering, STEM College, RMIT University, Melbourne, VIC 3001, Australia

**Keywords:** bitumen, asphalt, emissions, fumes, sample preparation, VOCs, PAHs

## Abstract

Bitumen is a major construction material that can emit harmful fumes when heated. These fumes pose health risks to workers and communities near construction projects or asphalt mixing plants. The chemical complexity of bitumen fumes and the increasing use of additives add to the difficulty of analytically quantifying the harmful chemicals emitted using a single technique. Research on bitumen emissions consists of numerous sample preparation and analytical methods. There are a range of considerations to be made when deciding on an appropriate sample preparation method and instrumental configuration to optimise the analysis of specific organic contaminants in emissions. Researchers investigating emissions from bituminous materials may need to consider a range of analytical techniques to quantify harmful chemicals and assess the efficacy of new additives. This review summarises the primary methodologies for sample preparation and analytical techniques used in bitumen research and discusses future challenges and solutions.

## 1. Introduction

Bitumen is a product produced from the refining of crude oil using vacuum distillation. The resulting bitumen is used as an industrial binder in a range of road construction; this type of refined bitumen has been used since the early 1900s [1,2]. Throughout this article, bitumen and asphalt are referred to as specific terms. When applied in paving construction, bitumen is defined as a binder to hold aggregate together. Aggregate often refers to a mass of crushed rock or similar material. This bitumen–aggregate mixture is referred to as asphalt, a viscoelastic material that can be mixed and laid using special machinery when hot. When the material is cooled it sets and becomes much harder. This asphalt is what constitutes road and airport pavements [3].

In a report by Eurobitume and the Asphalt Institute, paving projects account for 85% of bitumen produced, 10% in roofing applications, and the remainder used in various smaller applications such as waterproofing [4]. As it is part of the heavy residue resulting from distilled crude oil [3], the composition of bitumen is a complex mixture of organic compounds that varies depending on the crude oil source and the manufacturing methods used at the refinery [3,5].

At room temperature, bitumen is still considered a liquid but is solid for all practical purposes. Due to the material’s high viscosity, high temperatures are used when working with it to allow mixing with the aggregate. When used in paving applications the working temperatures can vary widely. There are four commonly defined types of asphalt relating to operation temperature: hot mix asphalt (HMA) (150–190 °C), warm mix asphalt (WMA) (100–140 °C), half-warm mix asphalt (HWMA) (60–100 °C), and cold mixes (0–40 °C) [6,7,8,9]. The mixing and laying temperatures can also go as high as 200 °C when bitumen is overheated in certain situations or during spray sealing operations [10]. When the bitumen is heated, particularly to the upper temperature ranges, it releases fumes, the volume of these fumes is reported to increase with temperature [11,12,13,14,15,16,17,18]. These emissions contain an extensive range of organic compounds. Among these emissions are volatile and semi-volatile organic compounds (VOCs/SVOCs); polyaromatic hydrocarbons (PAHs); and heterocyclic compounds containing nitrogen, oxygen, or sulfur, as well as inorganic gases such as hydrogen sulfide [7,8,10,12,19,20,21,22,23,24,25,26]. These are broadly categorised in Table 1. PAHs comprise a large proportion of the chemical classes investigated in laboratory-generated fumes due to their carcinogenicity [27] (Table 1). However, compounds from various chemical classes besides PAHs may also be of concern given their adverse health effects [28,29]. In particular, benzene is poorly reported in the literature relative to PAHs but is a known carcinogen reported in bitumen emissions [20,30,31]. Other simple arenes known to be contaminants in bitumen are also poorly reported in this field, but they are commonly screened as environmental contaminants due to their serious health concerns [20,30,32].

## 2. Regulations and Past Research

A report produced by National Institute for Occupational Safety and Health (NIOSH) in 2000, drawing upon research from the 1970s onward, concluded that the information at the time was insufficient to quantifiably measure the health impacts of exposure to bitumen fumes [39]. Since then, the research on bitumen fumes has been progressively expanded, and in 2011 the world health organisation (WHO) and the IARC (International Agency for Research on Cancer) labelled straight run bitumen used in paving as “possibly carcinogenic to humans” (group 2B) [40].

While research into the carcinogenicity of bitumen has continued, understanding the chemical composition of bitumen fumes and how these are released still appears to require further investigation. The current regulatory method provided by the NIOSH method 5042 (issue 1, 1998) for measuring total particulate matter and polyaromatics as benzene soluble matter [41]. Method 5042 requires PTFE filters attached to pumps that can control the air flow, the pumps are run for a set time to standardise the volume that moves through them. Once the set volume of air has moved through the filters, they are weighed to measure total particulate matter (TOM) and then extracted with solvents to measure PAH content gravimetrically as benzene soluble matter (BSM) or solvent soluble matter (SSM) if using anything other than benzene [42]. This measurement is converted to mass/volume allowing an estimation of the concentration of particulates and aromatics in the air at worksites or from laboratory samples. The gravimetric analysis is a simple quantitative method, and the simplicity of this technique allows for quick testing that does not require expensive instruments. Therefore, it is highly accessible and similar methods have been adopted in many countries as a regulatory method [43,44,45,46].

However, the drawback of this method in a research setting is the lack of any identification of specific chemical species present in the sample, such that two samples of the same concentration may pose different levels of harm to humans and the environment depending on the concentration of the chemicals present [27,35,47]. To address this, researchers commonly employ different methods for laboratory research on fuming [23]. Some of the analytical procedures that are commonly used when investigating bitumen emissions are generalised analytical methods for VOCs or PAHs, such as the NIOSH method 5515 for PAHs, and while they can identify VOCs and PAHs, they do not provide adequate information for sample preparation specific to bitumen [27,48]. Due to the lack of agreement on standardised methods and some uncertainty regarding which chemicals are most prevalent or essential to monitor, researchers have many options when planning an investigation on bitumen fumes. While many compounds have general exposure limits attached to them, which apply in asphalt paving worksites where compounds such as hydrogen sulfide, naphthalene, and other VOCs are present, there are currently no internationally standardised methods for determining specific chemicals in bitumen fumes. Further, specific regulations in a country or state can vary significantly making them inconsistent in a global setting [4].

Bitumen fumes are a complex mixture of chemicals, containing over 100 different organic species in the emissions alone [16]. Although there are measures in place to ensure worker and environmental safety is upheld, there is a general lack of understanding about the full impact of exposure to bitumen fumes [49]. Further, the composition and release of asphalt fumes are not clearly understood [16,50]. This means there may also be other chemicals of concern, such as sulfur-containing polyaromatic hydrocarbons (SPAHs) or particular VOCs that are not always considered in analytical assessments and could contribute to the overall danger of bitumen emissions to the environment and humans [20,51].

This review will cover only bitumen sample preparation techniques that have been used in controlled laboratory settings and reported in the literature not those that are collected in the field testing of construction sites. These field-testing methods are detailed elsewhere in the literature [8,21,45,52,53,54].

## 3. Sample Collection and Sample Enrichment Methods

Research investigating fuming can either be performed in the field, where samples of fumes are taken on a work site to measure worker exposure [8], or sampling can be performed in the laboratory in a closed system as a controlled experiment [9]. When focusing on the laboratory setting, there is a range of methods to assess fumes accurately. Each sample preparation method seeks to collect the chemical components of the fumes before moving them to an analytical instrument for identification. The methods of collection that have currently been used in this area of research can be split into three main categories: filters [7,8,9,17,33,35,38,48,52,55,56,57,58], headspace sampling [20,59,60], and direct evolution of fumes into analytical instruments [19,30,56,61]. Each of these techniques has unique advantages and disadvantages.

### 3.1. Filter Sampling

The collection of fumes via filtering is a commonly used technique in the laboratory [7,8,9,17,33,35,38,48,52,55,56,57,58]. The use of filters allows for large volumes of bitumen fumes to be concentrated onto a series of filters significantly increasing the concentration of trace compounds that are able to be collected. Once the collected analytes are extracted with a solvent, the extract can then be separated using gas or liquid chromatographic (LC) instruments before being identified with methods such as mass spectrometry (MS) [62].

When using filters in the laboratory, a mixing apparatus is often used to stir while heating the bitumen (Figure 1). The agitation and heating are intended to mimic in-field conditions and facilitate the release of particulates [9]. However, it is not a complete simulation of such conditions, and reproducibility is difficult because factors such as container size, bitumen surface area, agitation, air flow, wind, environmental temperature, etc. can affect the results [9,16].

The gaseous emissions from bitumen during heating and agitation are pumped through tubing into filters to collect the sample. Various configurations of this system are used throughout the research and there is no universally standardised system [7,8,9,17,33,35,38,52,55,57,62,63,64]. In general, the system consists of a PTFE or glass wool filter to physically remove particulates and downstream of this there is at least one filter to collect the gaseous chemicals; this is often a charcoal filter for VOCs and an XAD-2 filter for PAHs. Often a combination of both filters is used to analyse a broader range of chemicals in the emissions [9,63]. Once the emissions have been collected, the filters can be extracted with minimal amounts of solvent to produce a concentrated extraction of the sample. This extract can then be separated using a gas chromatogram (GC) or LC techniques and identified using a range of detectors such as a MS [35].

### 3.2. Headspace Sampling

Headspace (HS) sampling can generally be categorised into static headspace (SHS) or dynamic headspace (DHS) sampling. SHS sampling involves using a speciality syringe to draw a gas sample from the HS of a vial containing a small mass of the sample, which is then injected into a GC.

SHS is a non-exhaustive extraction and takes a small sample of the gas phase with a gas tight syringe. This sample is from a single point in time and the analytes extracted will represent the equilibrium between the sample matrix and the HS when the extraction occurs. Taking an extraction while the HS is not at equilibrium can lead to inconsistent results [65].

For DHS sampling, instead of taking a single sample of gas from the HS of a vial, the gases are continuously extracted along with a carrier gas through capillaries to the GC. The gases are often concentrated onto a sorbent trap to allow detection of trace level gases. This concentration of analytes allows for much greater detection of chemicals that would be in too low of a concentration in a single small volume sample. This method is an exhaustive technique that depletes the sample during extraction [66].

Research by Boczkaj et al. [20] uses both an SHS and DHS sampling method to investigate bitumen fumes. The SHS method took a 0.5 mL vapour sample from a 22 mL vial containing 0.1 g of bitumen which is heated to 180 °C and held for 30 min before injecting the gas into a GC for separation and analysis. Using SHS, the researchers analysed sulfur containing compounds by applying a pulsed flame photometric detector (PFPD), and nitrogen containing compounds using a nitrogen phosphorus detector (NPD) and flame ionisation detection (FID). The use of PFPD and NPD on sulfur and nitrogen containing compounds, respectively, allowed for the accurate trace level detection of these compounds in a complex mixture of volatile compounds. The DHS method by the researchers used the same vial, sample weight and preparation technique. The HS gases were moved from the heated sample into a sorbent trap which was held at 30 °C. The analytes were then desorbed from the trap and transferred directly into a GC by heating the trap to 270 °C. The compounds were identified by using mass spectrometry as the detector. Using both SHS- and DHS-GC with MS, FID, PFPD, and NPD, the researchers successfully identified a range of VOCs containing nitrogen and sulfur such as pyridine, carbon disulfide and hydrogen sulfide.

This method is advantageous when examining the most volatile fraction of bitumen fumes. Notably, hydrogen sulfide and carbon disulfide, which were measured in the fumes by the researchers. The limits of these HS techniques are determined by larger compounds possessing low volatility as these compounds tend to be emitted with particulate matter generated through high temperatures and agitation [50].

### 3.3. Headspace Solid Phase Microextraction Sampling

Headspace solid phase microextraction (HS-SPME) is a non-exhaustive technique similar in principle to the previous example of HS; however, instead of taking a gas sample of the HS a SPME device is exposed to the gaseous analytes in the headspace where the chemicals will absorb into the solid phase of the device. The device itself consists of a thin silica needle-like structure covered in a solid phase coating (Figure 2). An example of this coating is the nonpolar coating, polydimethylsiloxane, which is one of the most common coating materials utilised in HS-SPME. The coating allows chemicals in the HS to be concentrated onto the fibre; the fibre is then removed from the headspace and inserted into the inlet linear to the GC. The analytes are then desorbed in the heated liner and the gas flow carries the analytes into the column [66].

To quantify the analytes, HS-SPME relies on the equilibrium of a compound between the sample matrix, HS, and the fibre. Quantification methods that rely on this equilibrium relate to the concentration of analyte in the sample, so analyte concentration is generally measured in weight/weight (*w*/*w*). In comparison, other methods for quantifying specific analytes in bituminous material often require organic extractions and separations to obtain the chemical of interest from the bitumen sample; however, this is time consuming and produces waste from the organic solvents used [35].

Research conducted by Tang and Isacsson [60] demonstrated a successful HS-SPME method for quantifying BTEX (benzene, ethylbenzene, toluene, and xylenes) in bituminous emulsions using GC-MS. The researchers developed a method using a 100 µm diameter polydimethylsiloxane SPME fibre. An internal standard addition method was used to quantify the compounds, and an external calibration method using a surrogate sample matrix was used and compared. The researchers successfully quantified the selected volatiles in the low parts per million weight (ppmw) ranges using external calibration and internal standard addition methods. The authors did note that the internal standard method performed better and that an internal isotopically labelled standard may be the better approach. One of the advantages of this method is the ability to detect trace level VOCs such as benzene and xylene, which can be challenging to extract from bituminous material. These compounds are seldom reported in the literature on this topic [60]. The method used is also quick and reproducible. It does not rely on solvents or clean up procedures reducing the solvent waste produced.

### 3.4. Nonseparative Real Time Sampling

These methods involve the continuous flow of gases from the sample into an analytical instrument. The gases are passed from the sample container into an instrument. This provides the advantage of detecting any changes in the chemicals being emitted from the sample over time or varying temperature ranges.

Work by Cui et al. [61] employed this analytical technique using a thermogravimetric analyser (TGA) coupled with MS to continuously qualitatively analyse the emissions produced by bitumen samples with varying concentrations of layered double hydroxide additives to investigate the effect of this additive on the emission of volatile species. The TGA records the mass loss by using a temperature program that increases temperature over time at a rate of 10 °C/min from room temperature to 300 °C. Piping carries gasses from the TGA to the MS, where selected molecular ions of VOCs are monitored. The intensity of the response for each molecular ion can be used to determine the relative abundance of the molecule passing the detector.

The advantages of this method allow for the identification of a wide range of compounds. In this work the researchers can identify small volatiles including hydrogen sulfide and larger PAHs such as fluoranthene. The real-time monitoring of sample emissions means that the variation in fume composition can be tracked over time or with temperature changes. However, the procedure described by the researchers does not quantify the chemical concentrations. The results are described using current intensity measured in amperes (A). The researchers use this as a qualitative measure of volatile emissions in different bitumen samples.

Research by Borinelli et al. [30] employed a continuous sampling method to monitor a range of VOCs using the proton transfer reaction time of flight mass spectrometry (PTR-TOF-MS). Their method involved heating 5 g samples of bitumen in a sealed container and the evolved gases were transferred via capillaries to a PTR-TOF-MS instrument to be identified. Using this method, the researchers identified a wide range of VOCs categorised into alkanes, aromatics, and sulfur containing VOCs.

In this work the investigators were able to measure analyte response over a range of temperatures in real time. The researchers were able to quantify concentrations of specific chemicals and their classes in terms of emission rates (µmol m^−2^ S^−1^) with good sensitivity. Their results showed differences in emission rates between modified and unmodified bitumen and an increase in the monitored VOC emissions as temperature increased.

### 3.5. Summary of Sample Preparation Techniques

The sample preparation methods available based on the current research provide a range of tools for researchers when analysing bitumen emissions. When designing an analytical analysis, selecting an appropriate sample preparation method is vital if working with a complex material such as bitumen. Due to the range of organic compounds that are present in the emissions the choice of whether to investigate acetone [30] or benzo[a]anthracene [33] would reasonably require different analytical procedures. There are both advantages and limitations to the methods described in this work. Some of these are outlined in Table 2.

Historical techniques, such as filter sampling, have proved essential in analysing trace PAHs in bitumen due to the concentration of large emission volumes onto filters [9,38]. However, this method requires solvent extraction of filters, large sample volumes, and a timely extraction process involving ad hoc mixing apparatuses. Future technologies such as novel SPME materials designed for selective adsorption of aromatic compounds could prove crucial for analysing trace PAHs in bitumen using fast, automated methods that have become a mainstay in analytical chemistry [68,69,70,71].

## 4. Considerations for GC or LC Instruments and Detectors When Analysing Bitumen

Given the complex nature of bitumen fumes, GC and LC methods are often used to separate chemicals collected from the samples before they can be identified. Between the instruments, GC is the more commonly used throughout research on bitumen fumes [7,8,17,20,25,33,35,36,37,38,57,59,62,72]. The complex matrix of bitumen fumes and the chemical constituents’ high volatility and nonpolar nature suit GC instrumentation. The customisation of these instruments is extensive. Two notable points of customisation are detectors and columns. There is a range of detectors that have been used successfully in this area of research. The commonly available detectors (Table 3) can provide unique advantages when analysing specific classes of chemicals [66,73].

Columns can have a significant effect on the separation of compounds [73,74]. Different columns will generally be required for smaller VOCs, while larger PAHs containing more than three rings may require entirely different columns to reach a significant resolution of analyte peaks [73]. Examples of this can be seen in the columns employed in the EPA method 8260D [75] for VOCs and the EPA method 8275A for PAHs [76]. These are significantly different to allow for the separation of compounds with vastly different chemistry. The equipment described in method 8260D utilises a capillary column with a greater volume of stationary phase and small internal diameter better suited to separating low boiling point VOCs. In contrast, the 8275A method utilises a thinner stationary phase allowing higher boiling point compounds to be eluted in a reasonable time. Other considerations are the polarities of the target analytes as hydroxyl containing hydrocarbons may be more easily separated from the nonpolar gases of bitumen emissions using a column loaded with a polar stationary phase [74].

LC is less commonly used in bitumen fuming research. It has been applied successfully by measuring larger PAHs containing three or more rings, such as phenanthrene or benzo[a]pyrene, while employing a fluorescence detector [9,17,35,36,38,55,57,77]. This detector has the notable advantage of distinguishing between isomers of some PAHs [78]. This general method requires more sample preparation than many GC procedures as neither bitumen nor its emissions can be directly introduced into LC instruments. Instead, samples must be prepared in an appropriate solvent beforehand. This usually involves the concentrations of emissions or liquid–liquid/liquid–solid extractions of bitumen [9,17,35,36,38,55,57].

## 5. Concerning Additives for Fume Control in Bitumen

Multiple recent reviews covering many commonly used additives and their methods of action in detail are available [16,23,50]. These works cover a large extent of the current literature on additives in bitumen and only a small summary of additives will be covered here.

A significant assortment of additives can be added to bitumen to reduce emissions. Work by Wang et al. [23] provides a comprehensive review of bitumen additives and their work categories emission reducing additives into organic polymer materials, inorganic materials, and composite materials. The additives essentially function by retaining compounds within the bitumen that may otherwise become part of the fumes. In a review by Wang et al. [16] these mechanisms of fume suppression are broadly categorised into physical and chemical methods. The authors provide a more nuanced definition in their work. Although additives have been shown to reduce emissions in bitumen, it has been discussed that environmentally friendly emission reducing additives are important for the industry, given the current selection of commonly used additives that may produce emissions [23]. The expansion and standardisation of analytical procedures to measure the effectiveness of future additives are essential. Quantifying the emission reducing effects of an additive and identifying any emissions it may introduce will be essential to reduce environmental impacts and limit the health impacts on workers while potentially improving bitumen performance [16,23,50,79].

## 6. Conclusions

While a broad scope of analytical methods is available to investigate bitumen fuming, much of the research in this area uses similar instrumentation. Researchers may need to appropriately consider the aims of their analysis when selecting a sample preparation method and analytical technique as each method provides advantages and disadvantages. A single procedure detailed in this review cannot simultaneously identify and measure all analytes in a bitumen fume sample. A combination of methodologies is required to identify and quantify the range of chemical components present in bitumen. The ultimate challenge would be to develop methods using existing instrumentation or design new instruments which can detect all priority chemicals using a single technique or simplified procedure. These methods could include future iterations of SPME technologies or novel techniques. Achieving this may require a more robust definition of priority chemicals in bitumen emissions. Much of the current research focuses on PAHs. Moreover, while this class of compound is undoubtedly of concern, in comparison, arene compounds comprise smaller areas of this research field while still possessing significant health and environmental risks. The main chemical drivers of adverse health impacts from bitumen emissions should be clear so that analytical methods can more accurately report on the safety of bituminous materials.

## Figures and Tables

**Figure 1 molecules-27-05068-f001:**
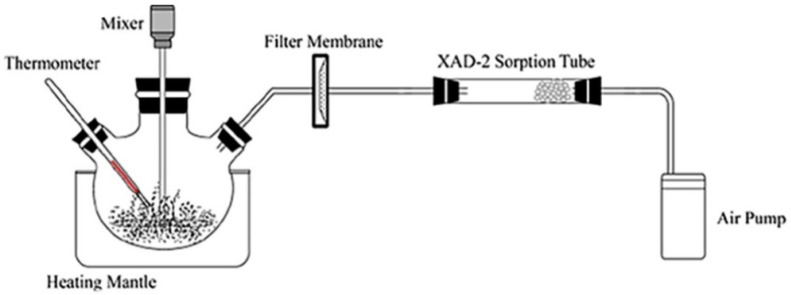
Diagram of mixing apparatus and filter system [18]. Reprinted from *Journal of Hazardous Materials*, vol 371, Shicong Mo, Yuhong Wang, Feng Xiong, Chunjin Ai, Effects of asphalt source and mixing temperature on the generated asphalt fumes, pg. 342–351, Copyright (2019), with permission from Elsevier.

**Figure 2 molecules-27-05068-f002:**
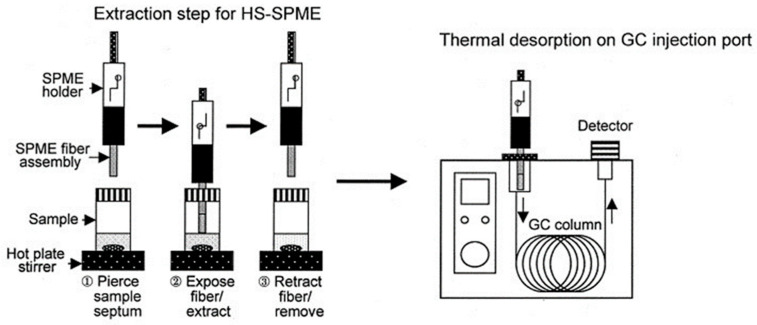
Diagram of HS-SPME device procedure [67]. Reprinted from *Journal of Chromatography A*, vol 880, Kataoka H, Lord LH, Pawliszyn J, Applications of solid phase microextraction in food analysis, pg. 35–62, Copyright (2000), with permission of Elsevier.

**Table 1 molecules-27-05068-t001:** List of broadly classified organic chemicals identified in bitumen fumes using laboratory testing.

Chemical Class	References
Polycyclic aromatic hydrocarbons	[7,8,17,33,34,35,36,37,38]
Nitrogen-containing polycyclic aromatic hydrocarbons	[34]
Oxygen containing polycyclic aromatic hydrocarbons	[37]
Sulfur-containing polycyclic aromatic hydrocarbons	[7,33,34,37]
Nitrogen-containing volatile organic compounds	[20]
Volatile organic compounds	[20,30]
Sulfur-containing volatile organic compounds	[20,30]

**Table 2 molecules-27-05068-t002:** Comparison of sample preparation techniques.

Sample Technique	Comment
Filter sampling	The most representative of industrial application Requires solvent extraction of analytes
Headspace sampling	Fast sampling Suited for large sample sets Solventless
Headspace solid-phase microextraction	Fast sampling Suited for large sample sets Solventless
Nonseparative real-time sampling	Solventless Possible to measures organic emissions rates over time

**Table 3 molecules-27-05068-t003:** Common detectors and uses [66].

GC Detectors	Comment
Mass spectrometer (MS)	Good for identifying many unknown compounds
Flame ionization detector (FID)	Good for quantitative analysis of known compounds
Flame photometric detector (FPD)	Highly sensitive for sulfur containing compounds
Nitrogen phosphorus detector (NPD)	Highly sensitive for nitrogen or phosphorous compounds

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
