# Peer review of "Sample Preparation and Analytical Methods for Identifying Organic Compounds in Bituminous Emissions"

_molecules, 2022, doi:10.3390/molecules27165068_

Round 1

Reviewer 1 Report

Minor revision needed

Abstract is not sufficient.

The authors should be added more sample collections methods.

Previous studies related to VOC and bituminous should be incorporated in the proper places.

·         10.1088/1752-7163/abf1d0

·         doi.org/10.3390/molecules26164948

·         https://doi.org/10.1016/j.jclepro.2022.131067

Author Response

Minor revision needed

Point 1: Abstract is not sufficient.

Response 1: We thank the reviewer for pointing this out. The abstract has been expended accordingly. Please see lines 10-20.

Point 2: The authors should be added more sample collections methods.

Response 2: We thank the reviewer for their suggestion. In this work we have intentionally limited the sample preparation methods to those which have been used for bitumen samples and not methods for sample preparation/collection from other fields. These have been recently reviewed for all methods that specifically use laboratory generated bitumen fumes.

Point 3: Previous studies related to VOC and bituminous should be incorporated in the proper places.

  • 10.1088/1752-7163/abf1d0

  • doi.org/10.3390/molecules26164948

  • https://doi.org/10.1016/j.jclepro.2022.131067

Response 3: We are thankful for the reviewer’s suggestions. The references provided by reviewer 1 have been considered. References 1 and 2 do not align with our current work as they lack specificity to bitumen. Reference 3 is relevant to this review and has been included (line 344).

Reviewer 2 Report

This paper presents a review on the testing of chemicals in bitumen fumes, which is very interesting and worth investigation. The paper needs some improvement for potential publication.

Please revise abstract. List the main methodology and findings from this paper in the abstract.

English expression must be improved.

Researchers might be interested in what type of chemicals are more harmful for health and the environment. Please elaborate.

Can you present a table listing the pros and cons of the sampling methods, such as in Table 2

Author Response

This paper presents a review on the testing of chemicals in bitumen fumes, which is very interesting and worth investigation. The paper needs some improvement for potential publication.

Point 1: Please revise abstract. List the main methodology and findings from this paper in the abstract.

Response 1: We thank the reviewer for pointing this out. The abstract has been revised and expanded. Please see lines 10-20.

Point 2: English expression must be improved.

Response 2: We thank the reviewer for this feedback. Grammar corrections have been made throughout the entire text.

Point 3: Researchers might be interested in what type of chemicals are more harmful for health and the environment. Please elaborate.

Response 3: We thank the reviewer for pointing this out. Some discussion regarding which chemicals in bitumen emissions may be of greater concern due to health and environmental impact have been added to the introduction and conclusion. Please see Lines 53-60 and lines 361-366.

Point 4: Can you present a table listing the pros and cons of the sampling methods, such as in Table 2

Response 4: We thank the reviewer for this suggestion. A section briefly summarising the sample preparation methods has been added to the end of this section (lines 266-281). A simple table has also been added that highlights important aspects of each specific method (line 286).

Round 2

Reviewer 2 Report

All comments are addressed.